# Multispectral Image Determination of Water Content in *Aquilaria sinensis* Based on Machine Learning

Peng Wang [1,2], Yi Wu [3], Xuefeng Wang [1,2,*], Mengmeng Shi [1,2], Xingjing Chen [1,2] and Ying Yuan [1,2]

1. Institute of Forest Resource Information Techniques, Chinese Academy of Forestry, Beijing 100091, China; sdwp2010@163.com (P.W.)
2. Key Laboratory of Forest Management and Growth Modelling, National Forestry and Grassland Administration, Beijing 100091, China
3. College of Forestry, Nanjing Forestry University, Nanjing 210037, China; wuyi_0416f@163.com
* Correspondence: xuefeng@ifrit.ac.cn

**Abstract:** The real-time nondestructive monitoring of plant water content can enable operators to understand the water demands of crops in a timely manner and provide a reliable basis for precise irrigation. In this study, a method for rapid estimation of water content in *Aquilaria sinensis* using multispectral imaging was proposed. First, image registration and segmentation were performed using the Fourier–Mellin transform (FFT) and the fuzzy local information c-means clustering algorithm (FLICM). Second, the spectral features (SFs), texture features (TFs), and comprehensive features (CFs) of the image were extracted. Third, using the eigenvectors of the SFs, TFs, and CFs as input, a random forest regression model for estimating the water content of *A. sinensis* was constructed, respectively. Finally, the monarch butterfly optimization (MBO), Harris hawks optimization (HHO), and sparrow search algorithm (SSA) were used to optimize all models to determine the best estimation model. The results showed that: (1) 60%–80% soil water content is the most suitable for *A. sinensis* growth. Compared with waterlogging, drought inhibited *A. sinensis* growth more significantly. (2) FMT + FLICM could achieve rapid segmentation of discrete *A. sinensis* multispectral images on the basis of guaranteed accuracy. (3) The prediction effect of TFs was basically the same as that of SFs, and the prediction effect of CFs was higher than that of SFs and TFs, but this difference would decrease with the optimization of the RFR model. (4) Among all models, SSA-RFR_CFs had the highest accuracy, with an $R^2$ of 0.8282. These results confirmed the feasibility and accuracy of applying multispectral imaging technology to estimate the water content of *A. sinensis* and provide a reference for the protection and cultivation of endangered precious tree species.

**Keywords:** *Aquilaria sinensis*; water content; multispectral image; feature extraction; machine learning

## 1. Introduction

*Aquilaria sinensis* is a tree species of *Aquilaria* in the family Thymelaeaceae and is mainly distributed in coastal provinces such as Hainan, Guangdong, and Guangxi in China and in some Southeast Asian countries [1]. The resin secreted by *A. sinensis* after injury has a strong fragrance and anti-inflammatory and antioxidant functions. It is widely used in the fields of incense making, pharmaceuticals, and handicraft manufacturing [2,3]. High-quality *A. sinensis* is very scarce and can be worth tens of thousands of dollars per kilogram. The preciousness of *A. sinensis* has caused wild *A. sinensis* to be overcut by humans, and coupled with habitat changes, natural wild *A. sinensis* is on the verge of extinction. In 2004, *A. sinensis* was officially listed in Appendix II of the Convention on International Trade in Endangered Species of Wild Fauna and Flora, to prohibit the illegal trade of *A. sinensis* and make it sustainable [4]. To protect this tree species and meet market demand, the planting of *A. sinensis* has been vigorously promoted in southern China. However, juvenile *A. sinensis* is very sensitive to water, and most operators cannot determine the water demand of

*A. sinensis* in real time, resulting in its poor growth and even death. Therefore, operators need a real-time and accurate non-destructive estimation method of the water content of *A. sinensis* to adjust the water conditions for the growth of *A. sinensis* in a timely manner, thus ensuring the quality and output of *A. sinensis*.

In actual production, an operator usually judges whether a plant lacks or has abundant water according to changes in its appearance and color, but this method largely depends on the actual experience and subjective judgement of the operator. In recent years, with the development of artificial intelligence and the Internet of Things, image-based plant water monitoring technology has gradually been applied in practice [5]. This approach has strong flexibility and operability. By establishing the relationship between spectral information and plant water content, real-time feedback on plant water demand can be realized in the subsequent operation process. Hyperspectral imaging technology can capture the rich spectral information of the measured plants and performs well in predicting plant water content [6,7]. For example, Yang et al. [8] used hyperspectral information to construct multiple Back Propagation Neural Network (BPNN) prediction models of winter wheat leaf water content after flooding eastern wheat, providing a theoretical basis for the prevention and control of winter wheat waterlogging disasters. Xuan et al. [9] used hyperspectral imaging technology to accurately evaluate the ripening period and water content of fresh okra fruit, providing technical support for farmers to optimize harvest dates. However, hyperspectral imaging technology still faces some problems, such as its high cost and excessive redundant information. In contrast, multispectral imaging technology does not have these problems, so it is easier to understand and apply [10,11].

At present, many scholars have applied multispectral imaging technology to plant water content monitoring. They have established the relationships between multispectral image features and plant water content by choosing different models to judge the water demands of plants. In previous studies, the most commonly used modeling methods were multivariate statistics and machine learning algorithms [12–14]. Among them, multivariate statistics can quantitatively describe the functional relationship between plant water content and various parameters, with stronger interpretability. However, the nonlinear mapping ability of machine learning algorithms is stronger than that of multivariate statistics. Models such as random forests, support vector machines, and neural networks are not sensitive to the absence of missing values (such as some attribute values in the sample) and they have strong anti-noise capabilities and good predictive capabilities [15].

When using multispectral images to estimate water content, the choice of explanatory variables is very important. Studies have shown that the green and near-infrared bands are ideal for identifying water in plant tissues, and related vegetation indices (VIs), such as the green normalized difference vegetation index (GNDVI), normalized difference vegetation index (NDVI), and optimized soil adjusted vegetation index (OSAVI), have been proven to be accurate in predicting plant water content [16,17]. For example, some scholars extracted multiple VIs from multispectral images of eggplant and built a linear water content prediction model one by one. The results showed that the two models with NDVI and OSAVI as independent variables had the highest prediction accuracy [12]. Torres et al. [13] used near-infrared imaging equipment to obtain the reflectance of 12 bands of olive tree multispectral images. After eliminating abnormal sample information through principal component analysis (PCA), partial least squares regression (PLSR) was used to construct the water content model of olive trees, and both the training and verification sets showed strong robustness, providing a reliable basis for operators to reasonably irrigate olive trees. Malvandi et al. [14] used the successive projection algorithm (SPA) to extract the reflectance of three characteristic bands from apple multispectral images and used them as explanatory variables to construct a PLSR prediction model for apple water content. The correlation coefficient ($R^2$) of the model reached 0.99, realizing the accurate and nondestructive detection of apple water content. In addition, some scholars used the texture features (TFs) of multispectral images as model parameters to estimate water content. For example, Zhou et al. [18] obtained canopy images of winter wheat by

using a UAV equipped with a multispectral sensor and constructed three wheat stomatal conductance estimation models, Cubist, BPNN, and Elaboration Likelihood Model (ELM), by extracting and combining the spectral features (SFs) and TFs of the image. The results showed that texture features were significantly correlated with wheat water content, and the accuracy of the comprehensive features (CFs) model based on TFs and SFs was more than 20% higher than that of the single-feature model.

The above studies have shown that multispectral images perform well in monitoring plant water content, but there are few reports on the estimation of water content in precious tree species. At the same time, most studies focus on image feature extraction and model construction, ignoring the process of image segmentation and model optimization. Based on the above considerations, the main goal of this study was to propose a multispectral image estimation method for the moisture content of *A. sinensis* seedlings under different moisture gradients and to analyze the effect of soil water content on the growth of *A. sinensis* seedlings. Based on the field water capacity, four water gradients were set up in the experiment, multispectral images of *A. sinensis* were obtained, and the moisture content was measured. On this basis, the water content estimation model of *A. sinensis* was constructed. The specific objectives were as follows: (i) Through the quantitative analysis of the growth difference of *A. sinensis* seedlings under different water gradients, clarify the specific influence of soil water content on its growth and determine the most suitable water conditions for the growth of *A. sinensis* seedlings. (ii) Combine image registration methods with segmentation algorithms to achieve the fast and accurate segmentation of *A. sinensis* multispectral images. (iii) Apply dimensionality reduction algorithm to eliminate multicollinearity of image features. On this basis, the water content of *A. sinensis* was predicted using SFs, TFs, and CFs to analyze the predictive performance of different image features. (iv) Apply the swarm intelligence optimization algorithm to adaptively optimize the model parameters in order to determine the best model for predicting the water content of *A. sinensis*.

## 2. Materials and Methods

The technical flow chart of this study is shown in Figure 1. Firstly, images of *A. sinensis* were collected using the Mica Sense Edge 3™ multispectral camera. Secondly, the Fourier–Mellin transform (FMT) was applied to register the images in different bands, and the fuzzy local information clustering algorithm (FLICM) was applied to segment the image, separating the foreground and background of the image. Again, spectral features (SFs) and texture features (TFs) in the foreground image were extracted, and composite features (CFs) composed of SFs and TFs were obtained. Then, the local tangent space arrangement (LTSA) was used to extract the feature vectors of the three types of image features, and the random forest regression model (RFR) for predicting the water content of *A. sinensis* was constructed, respectively. In the process of model construction, MBO, HHO, and SSA were used to adaptively optimize the numbers of decision trees and node features. Finally, the accuracy of the constructed model was tested to determine the optimal random forest regression model for predicting the water content of *A. sinensis*.

### 2.1. Overview of the Study Site

The study area was located in Wenchang City, Hainan Province, China (19°36′~20°3′ N, 109°12′~111°2′ E), with an average altitude of 42.55 m, belonging to the coastal plain on the northern edge of the tropics (Figure 2). There is no obvious seasonal variation in this area, the annual average temperature is 23.90 °C, the rainfall mainly occurs from May to October, the annual precipitation is 1721.60 mm, and the annual average humidity is 87%. The main soil is coastal sandy soil. The pH value of the soil is between 5.0 and 6.6, which is very suitable for the growth of tropical crops.

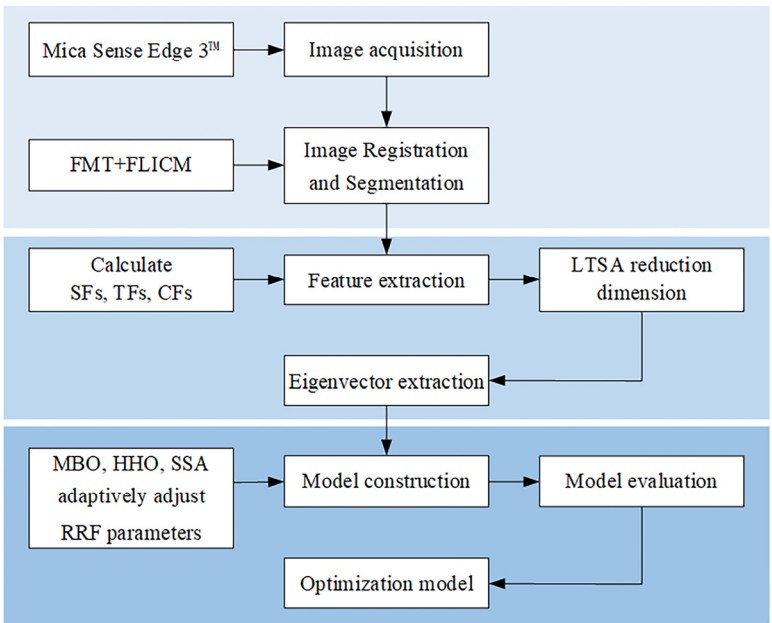

**Figure 1.** Technique flow chart.

**Figure 2.** Location of the study area.

### 2.2. Experimental Design and Data Acquisition

2.2.1. Experimental Design

We used seeds to raise seedlings in this experiment. After 2 years of growth, the tree height, crown width, and ground diameter of all *A. sinensis* seedlings were measured (mean values were 29.8 cm, 18.9 cm, and 6.1 mm, respectively). We selected 52 *A. sinensis* seedlings with no pests or diseases and uniform growth (The range of tree height was 29.8 ± 2 cm, the range of crown width was 18.9 ± 2 cm, and the range of ground diameter was 6.1 ± 1 mm) and moved them into flowerpots of the same size. Before transplanting the seedlings, 5 kg of air-dried seaside sandy loam was placed in each flowerpot and the soil nutrient content was measured using a soil nutrient meter, wherein the organic matter content was 10.5 g/kg, the available nitrogen content was 98.3 mg/ kg, the available phosphorus content was 3.38 mg/kg, and the available potassium content was 69.9 mg/kg. In this study, to simulate drought, flood, and normal conditions, we set 4 water gradients based on the field water-holding capacity (Table 1) with 13 replicates at each level, and we evenly divided 52 *A. sinensis* seedlings into 4 groups. To ensure the normal growth of *A. sinensis*, we applied the same amount of nitrogen, potassium, and phosphate fertilizers to the 4 groups during the experiment and carried out weeding and pesticide spraying. The experiment lasted for a total of 6 months. After the experiment, *A. sinensis* was moved indoors to take pictures and measure the basic growth factors and water content. On this basis, we used one-way analysis of variance to test the significance of the differences in height, crown growth, and ground diameter of *A. sinensis* under different water treatments and used Duncan's test to analyze the differences between groups. If there is a significant difference between the groups, it will be labeled with a different letter; if there is no significant difference between the groups, the same letter will be used for labeling.

**Table 1.** Soil water content under different water gradients.

| Category | Group | Soil Water Content |
|---|---|---|
| Control group | CK | 30%–40% |
| Treatment group 1 | T1 | 40%–60% |
| Treatment group 2 | T2 | 60%–80% |
| Treatment group 3 | T3 | 80%–90% |

2.2.2. Data Acquisition

In this study, we built a darkroom with a length of 1 m, width of 1 m, and height of 2 m using steel pipes and shading cloth (Figure 3). Except for the direction facing the camera, the other directions of the darkroom were covered by black shading cloth of aluminum foil composite film material. We installed 7 LED lights (Hangzhou SPL Photonics Co., Ltd., Hangzhou, China) on the top and front side frame of the darkroom as an active light source. *A. sinensis* was placed in the darkroom, and the camera was mounted using a tripod 2 m directly in front of the chamber. The camera center point and *A. sinensis* center point were kept at the same height. The camera used for image collection was a Mica Sense Ede 3™ equipped with 5 narrow-band spectral sensors, whose specific parameters are shown in Table 2. In actual operation, all *A. sinensis* were photographed according to the four orientations of due east, due west, due north, and due south. A total of 52 groups of 1040 images (1280 × 960 pixels) were obtained. After the images were taken, *A. sinensis* was cut at the base of the stem, and the fresh weights of the stem and leaf were determined using an electronic balance with an accuracy of 0.01 g. Then, it was dried in an oven (85 °C) and weighed. The water content was calculated using Equation (1):

$$WC = (FW - DW)/FW \times 100\% \tag{1}$$

where *WC* is the water content of *A. sinensis*, *FW* is the fresh weight, and *DW* is the dry weight.

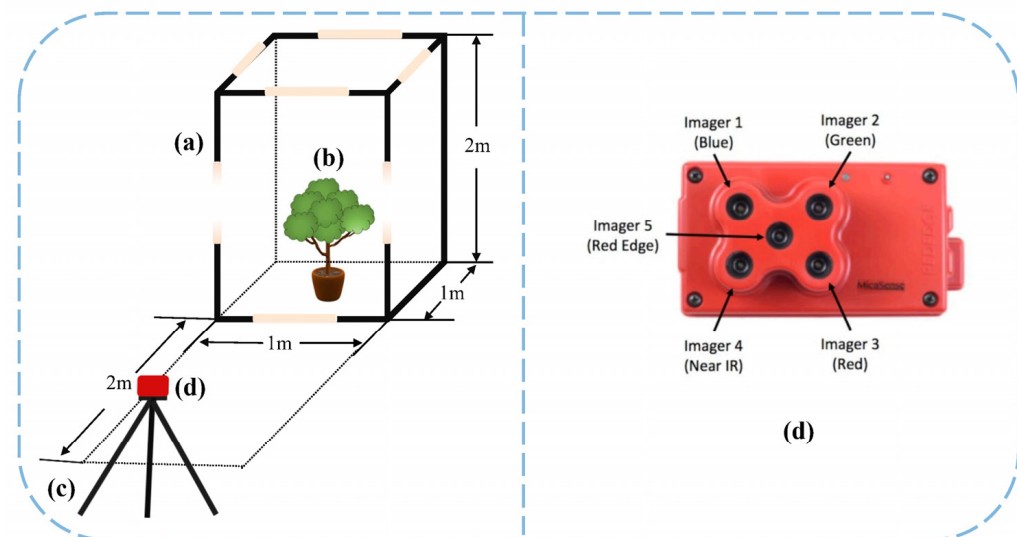

**Figure 3.** Schematic diagram of the image acquisition device: (**a**) darkroom; (**b**) *A. sinensis*; (**c**) tripod; (**d**) Mica Sense Edge 3™.

**Table 2.** Band information of Mica Sense Edge 3™.

| Band Name | Abbreviation | Center Wavelength (nm) | Bandwidth FWHM (nm) |
|:---:|:---:|:---:|:---:|
| Blue | B | 475 | 20 |
| Green | G | 560 | 20 |
| Red | R | 668 | 10 |
| Near IR | NIR | 840 | 40 |
| Red Edge | RE | 717 | 10 |

### 2.3. Image Processing

The multispectral images were acquired including a foreground region (called the region of interest, ROI) with *A. sinensis* information and a background region carrying a lot of irrelevant information. Before extracting the features of the *A. sinensis* multispectral image, it was necessary to segment the ROI area from the whole image. At the same time, the spectral sensors of the Mica Sense Edge 3™ used in this study were independent of each other, and there was a relative offset between the five discrete images obtained. If the *A. sinensis* image was directly segmented, the segmentation efficiency would be greatly reduced and the segmentation accuracy of each band would not be uniform. On the contrary, registering the images before segmentation could make the images of each band consistent in space, thus only the image with the highest definition and the easiest segmentation as the reference image for segmentation needed to be used, and the results could be directly applied to other bands.

#### 2.3.1. Image Registration

In this study, we applied the Fourier–Mellin Transform (FMT) to the registration of images. The FMT introduces polar coordinate transformation into the phase correlation method, which can realize fast registration of two images with rotation, zoom, and translation [19,20]. The principle of FMT is as follows.

Let two images be $f_1(x, y)$ and $f_2(x, y)$. Assume that there are rotation, scaling, and translation relationships between these images that can be expressed as shown in Equation (2):

$$f_2(x, y) = f_1[a(x\cos\theta_0 + y\sin\theta_0) - \Delta x, a(-x\cos\theta_0 + y\sin\theta_0) - \Delta y] \tag{2}$$

where $a$ is the scaling factor, $\theta_0$ is the rotation angle, and $\Delta x$ and $\Delta y$ are the displacements in the horizontal and vertical directions, respectively.

Fourier transforms are performed on $f_1(x, y)$ and $f_2(x, y)$, and the relative translation between the two images is obtained through the phase spectrum. The magnitude spectrum is calculated on both sides to obtain Equation (3):

$$M_2(u, v) = \frac{1}{a^2} M_1 \left[ \frac{1}{a}(u \cos \Delta\theta + v \sin \Delta\theta), \frac{1}{a}(-u \sin \Delta\theta + v \cos \Delta\theta) \right] \tag{3}$$

where $M_1(u, v)$ and $M_2(u, v)$ are the spectrum amplitudes of $f_1(x, y)$ and $f_2(x, y)$, respectively.

Since the spectrum amplitude is only related to the scaling factor and the rotation angle, the image is consistent with the scaling factor and the rotation angle of the spectrum amplitude. Therefore, the effects of rotation and scaling can be reduced by a logarithmic-polar coordinate transformation, and the polar coordinate transformation is shown in Equation (4):

$$M_2(\lg\rho, \theta) = \frac{1}{a^2} M_1(\lg\rho - \lg a, \theta - \Delta\theta) \tag{4}$$

where $M_1(\lg\rho, \theta)$ and $M_2(\lg\rho, \theta)$ are the logarithmic polar coordinates of $M_1(u, v)$ and $M_2(u, v)$, respectively.

In polar coordinates, the rotation and scaling of the two images are transformed into translation $(\lg\rho, \theta)$. At this time, the relative displacement of the two rotated, scaled, and translated images can be calculated by using the phase correlation method again, and the registration between images can be realized.

### 2.3.2. Image Segmentation

Affected by factors such as the sensor material and working environment, noise will inevitably be introduced when collecting multispectral images, and it is difficult to accurately segment noise using traditional segmentation algorithms. To meet the research needs, we applied the fuzzy local information clustering algorithm (FLICM) with a strong anti-noise ability for image segmentation. To evaluate the image segmentation effect, we used the partition coefficient $Vpc$ and partition entropy $Vpe$ as the evaluation indices. The calculation methods of $Vpc$ and $Vpe$ are shown in Equations (5) and (6), respectively.

$$Vpc = \sum_{i=1}^{N} \sum_{k=1}^{K} u_{ki}^2 / N \tag{5}$$

$$Vpe = -\sum_{i=1}^{N} \sum_{k=1}^{K} u_{ki} \times \log(u_{ki}) / N \tag{6}$$

where $N$ is the total number of pixels, $K$ is the number of clusters, and $u_{ki}$ is the membership degree of the pixel belonging to the Kth class.

### 2.4. Feature Extraction

After the image segmentation was complete, we extracted the gray features of the image and obtained the reflectance of the B, G, R, NIR, and RE bands by calculating the gray mean value of each band image in the four directions. On this basis, 20 VIs directly or indirectly related to the water content were extracted by combining the reflectivity of each band [21–23], and 25 spectral signatures (SFs) composed of spectral reflectance and VIs were finally obtained, as shown in Table 3.

**Table 3.** Vegetation indices related to water content.

| Vegetation Index | Abbreviation | Formula * | Reference |
|---|---|---|---|
| Difference Vegetation Index | DVI | NIR − R | [24] |
| Normalized Difference Vegetation Index | NDVI | (NIR − R)/(NIR + R) | [25] |
| Excess Green Index | EXG | 2 × G − R − B | [26] |
| Normalized Green-Bule Difference Index | NGBDI | (G − B)/(G + B) | [26] |
| Normalized Green-Red Difference Index | NGRDI | (G − R)/(G + R) | [26] |
| Water Index | WI | (G − B)/(R − G) | [26] |
| Renormalized Difference Vegetation Index | RDVI | NIR − R/(NIR + R)1/2 | [27] |
| Green Normalized Difference Vegetation Index | GNDVI | (NIR − G)/(NIR + G) | [28] |
| Modified Simple Ratio | MSR | (NIR/R − 1)/(NIR/R + 1)1/2 | [29] |
| Normalized Difference Water Index | NDWI | G − NIR/G + NIR | [30] |
| Enhanced Vegetation Index | EVI | 2.5 × (NIR − R)/(NIR + 6 × R − 7.5 × B + 1) | [31] |
| Kawasaki Index | IKAW | (R + B)/(R − B) | [32] |
| Simple Ratio Vegetation Index | SR | R/G × NIR | [33] |
| Chlorophyll Index | CI | (R + G)/2 | [34] |
| Red-Edge Chlorophyll Vegetation Index | RECI | (NIR/RED) − 1 | [35] |
| Excess Red Index | EXR | 1.4 × R − G | [36] |
| Excess Green Minus Excess Red | EXGR | 3 × G − 2.4 × R − B | [36] |
| Normalized Difference Red Edge Vegetation Index | NDRE | (NIR − RE)/(NIR + RE) | [37] |
| Red Green Blue Vegetation Indices | RGBVI | (G$^2$ − B × R$^2$)/(G$^2$ + B * R$^2$) | [38] |
| Green Index | GLI | (2 × G − R − B)/(R + G + B) | [39] |

* Among them, B, G, R, NIR, and RE are the pixel gray mean of the foreground image.

Texture features can effectively reflect the representational changes of *A. sinensis*, so we described the texture features by extracting the grayscale cooccurrence matrix (GLCM) of the image. In this study, the GLCM of the image was extracted from four directions, 0°, 45°, 90°, and 135°, and the image contrast (*CON*), correlation (*COR*), angular second-order moment (*ASM*), inverse differential moment (*IDM*), and entropy (*ENT*) were calculated in the four directions. To reflect the grayscale transformation of the image in different directions, we took the mean values of *CON*, *COR*, *ASM*, *IDM*, and *ENT* in the four directions as the texture features of the image, and the calculation methods are shown in Equations (7)–(11). There were 5 features per band, so a total of 25 texture features (TFs) were extracted. For ease of distinction, we named the extracted texture features in the form of X_Y, e.g., B_CON, which represents the contrast of B-band images.

$$CON = \sum_i \sum_j (i - j)^2 P(i, j) \tag{7}$$

$$COR = \left[ \sum_i \sum_j ((ij)P(i,j)) - \mu_x \mu_y \right] / \sigma_x \sigma_y \tag{8}$$

$$ASM = \sum_i \sum_j P(i,j)^2 \tag{9}$$

$$IDM = \sum_i \sum_j \frac{P(i,j)}{1 + (i-j)^2} \tag{10}$$

$$ENT = -\sum_i \sum_j P(i,j) \log P(i,j) \tag{11}$$

where $i$ and $j$ are image gray levels, $P(i, j)$ is the probability of $i$ and adjacent $j$ gray level, $\mu_x$ and $\mu_y$ are mean values, and $\sigma_x$ and $\sigma_y$ are standard deviations.

After extracting the SFs and TFs, they were combined to obtain the composite features (CFs) containing 25 spectral features and 25 texture features.

*2.5. Data Analysis and Modelling*

2.5.1. Data Dimensionality Reduction

When the feature vector has multicollinearity, it will lead to overfitting of the model. If the data are reduced at this time, this problem can be effectively solved. In this study, we used three dimensionality reduction algorithms, linear discriminant analysis (LDA), local tangent space arrangement (LTSA), and maximum variance spread (MUV), to extract the feature vectors of the SFs, TFs, and CFs and to construct the models. In this study, we used the intrinsic_dim function in drtoolbox to estimate the intrinsic dimensions of the SFs, TFs, and CFs (the minimum number of dimensions required to solve high-dimensional optimization problems), which were 2, 2, and 3, respectively, that is, from 25, 25, and 50 dimensions to 2, 2, and 3 dimensions, respectively. Among them, LDA is similar to PCA, which applies the idea of matrix decomposition in dimensionality reduction and can map the initial sample to a lower dimension sample space. However, unlike PCA, LDA is a supervised linear dimensionality reduction algorithm, which can ensure that the mean difference between various types of data is the largest and the intra-class variance is the smallest. LTSA is an unsupervised nonlinear manifold learning algorithm. It can calculate the overall low-dimensional embedding coordinates by rearranging the projection coordinates of the local space to achieve dimensionality reduction. MUV is similar to LTSA, but MUV is a global algorithm that not only considers the local information of the sample but also considers the relationship between the sample points and the non-adjacent sample points, thereby extending the high-dimensional data manifold in the low-latitude space.

The three algorithms were selected by comparing the accuracy of the models. The constructed model was named in the form of X-Y_Z, where X was the dimensionality reduction algorithm, Y was a regression model, Z was the feature type, and X-Y-Z was a subset of Y-Z. For example, LDA-RFR_SFs was a subset of RFR_SFs, which represented a random forest regression model that took the spectral features after dimensionality reduction by the LDA algorithm as input.

2.5.2. Random Forest Regression Model

Random forest (RF), a supervised ensemble learning algorithm based on decision trees that can be used for both classification and regression, is one of the most practical algorithms in bagging ensemble strategies [40]. The random forest regression (RFR) model consists of multiple decision trees. Each tree is independent of the others and does not affect the other, thus the final result of the model is jointly determined by each decision tree [41].

2.5.3. Model Parameter Optimization

When the number of decision trees is too large, it may lead to overfitting of the model. On the contrary, when the number of decision trees is too small, it might not ensure the fitting accuracy of the model. In addition, the number of randomly sampled variables (number of node features) when building a decision tree branch will also affect the accuracy of the model. This is because the number of node features controls the degree of randomness introduction. When the number of features extracted by tree nodes is large, the strength of each decision tree increases, but the overall randomness decreases. When the number of features extracted by the tree nodes is small, the overall randomness increases, but the complexity of each tree decreases. Applying the intelligent optimization algorithm to adaptively optimize the model parameters can quickly find an optimal parameter eigenvector, so that the model fitting effect can be optimal, and the RMSE of the model is used as the fitness function in the optimization process, which can also avoid overfitting of the model. Based on the above considerations, this study used three population optimization algorithms, the monarch butterfly optimization (MBO) [42], Harris hawks optimization (HHO) [43], sparrow search algorithm (SSA) [44], to adaptively optimize the numbers of decision trees and node features in the RFR model. The naming rules referred to those in Section 2.5.1 where X represented the optimization algorithm, such as MBO-RFR_SFs.

2.5.4. Model Evaluation

Due to the small number of samples (52), this study adopted the leave-one-out cross-validation method (LOOCV) to train the model, that is, one different sample was drawn from 52 samples each time as the test set, and the remaining 51 samples were used as the training set. By analogy, training 52 times so that all samples could be used as a test set, a prediction model containing 52 sub-models was obtained. The accuracy of the model was evaluated by the correlation coefficient ($R^2$), root mean square error (*RMSE*), and mean absolute percentage error (*MAPE*), and the calculation methods were as follows:

$$R^2 = 1 - \frac{\sum\limits_{i=1}^{n}(y_i - \hat{y}_i)^2}{\sum\limits_{i=1}^{n}(y_i - \overline{y})^2} \tag{12}$$

$$RMSE = \sqrt{\frac{1}{n}\sum_{i=1}^{n}(y_i - \hat{y}_i)^2} \tag{13}$$

$$MAPE = \frac{1}{n}\sum_{i=1}^{n}\left|\frac{\hat{y}_i - y_i}{y_i}\right| \times 100\% \tag{14}$$

where $n$ is the number of samples, $y_i$ is the measured value, $\overline{y}$ is the average of all, and $\hat{y}_i$ is the predicted value.

## 3. Results

### 3.1. Effect of Soil Water Content on the Growth of A. sinensis

A shown in Figure 4a,b, there were significant differences in tree height and canopy growth under different water treatments ($p < 0.05$). The tree height and crown width of the T1 group increased by 17.0 and 11.2 mm, respectively, and the tree height and crown width of the T2 group increased by 11.9 and 5.8 mm, respectively, which were significantly higher than those of the CK group. However, the tree height and crown width of the T3 group increased by only 3.1 and 2.3 mm, respectively, which were significantly less than the T1, and T2 groups. This showed that both drought and waterlogging conditions could inhibit the growth of *A. sinensis*, but the inhibitory effect of the drought condition was more obvious. As shown in Figure 4c, the difference in ground diameter growth under different water treatments was not significant ($p < 0.05$). This showed that the effect of water treatment on ground diameter was relatively small compared with tree height and crown width. Compared with the other treatment groups, the tree height, crown width, and ground diameter growth of the T3 group were the largest, and the differences with the CK group were the most significant, indicating that a soil water content of 60%–80% was most suitable for the growth of *A. sinensis*.

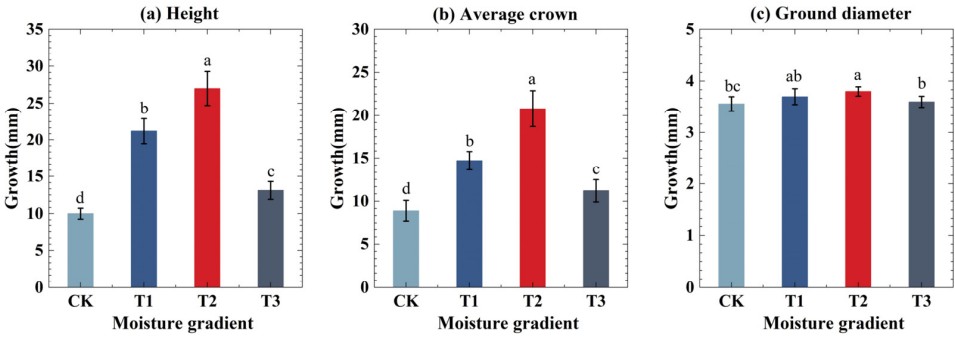

**Figure 4.** The growth differences of height (**a**), ground diameter (**b**), and crown width (**c**) of *A. sinensis* under different water gradients. The letter is used to mark whether the difference between the groups is significant.

### 3.2. Image Segmentation Effect

In the initial visual interpretation of the raw multispectral images, the B-band image with less noise and obvious foreground background difference was selected as the reference image, and the remaining band images were registered. The results are shown in Figure 5a. After the B-band image was segmented, a binary image was obtained, which was used as a mask to perform point multiplication with the five-band images at the same time to obtain the final segmentation result, as shown in Figure 5b. Among them, B, G, R, RE, and NIR represent the images of each band, and RGB represents the fused R, G, and B band images. Figure 5a shows that the RGB image synthesized after registration had no band shift and no band information was missing. Figure 5b shows that the synthesized RGB image was consistent with the foreground in Figure 5a, and the information was effectively retained. This study evaluated the effect of image segmentation more accurately by calculating $Vpc$ and $Vpe$. The larger the $Vpc$ value and the smaller the $Vpe$ value, the smaller the fuzziness of the segmentation matrix, the more accurate the pixel classification, and the better the segmentation effect. Table 4 shows the values of $Vpc$ and $Vpe$ under the optimal and worst segmentation effects. The results showed that after using FLICM to segment all B-band images, the $Vpc$ value of the segmentation matrix was be-tween 0.9640 and 0.9771 and the value of $Vpe$ was between 0.0233 and 0.0344, indicating that the model degree of the segmentation matrix was lower than 3.60%, the classification accuracy of the image pixels was higher than 97.67%, and the blurring degree of the segmentation matrix was very small. Only a few pixels were misclassified and the segmentation worked well. Figure 5 and Table 4 show that the application of FMT + FLICM could achieve fast and accurate segmentation of multispectral images.

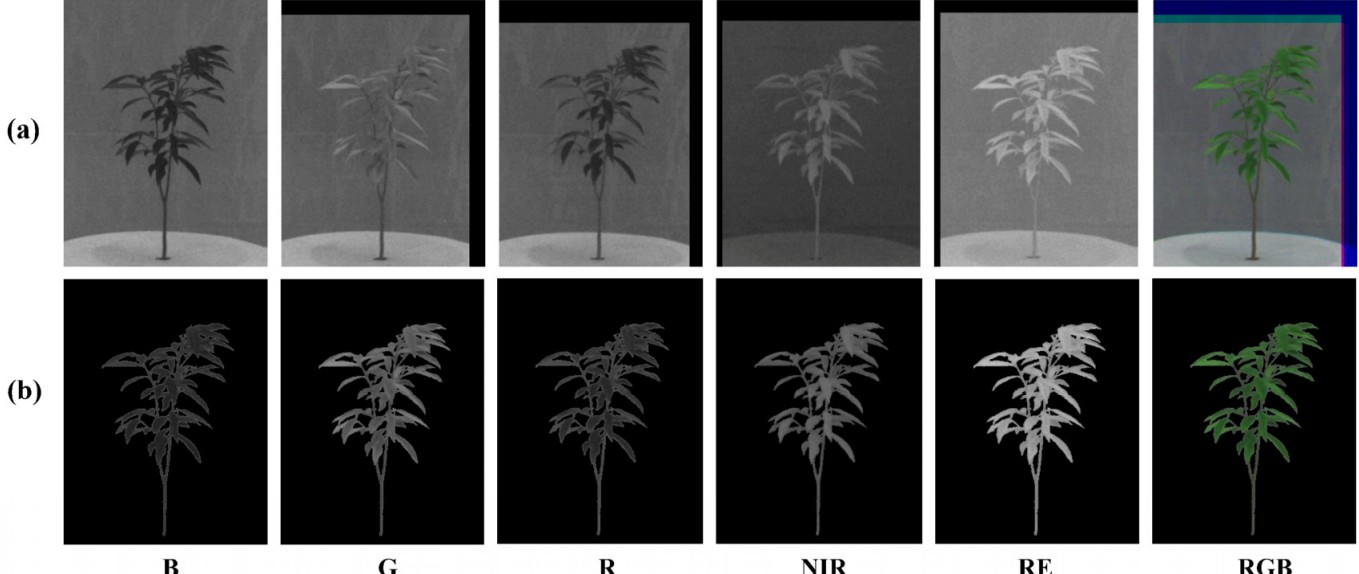

**Figure 5.** Image registration (**a**) and segmentation (**b**) results.

**Table 4.** Image segmentation effect evaluation.

| Evaluation Index | Best Effect | Worst Effect | Average |
|:---:|:---:|:---:|:---:|
| $Vpc$ | 0.9771 | 0.9640 | 0.9722 |
| $Vpe$ | 0.0233 | 0.0344 | 0.0262 |

### 3.3. A. sinensis Water Content Prediction Model

#### 3.3.1. Correlation Analysis

The SFs and TFs together form the CFs. As long as one of the SFs or TFs has multi-collinearity, the CFs need dimensionality reduction. Therefore, we performed correlation

analysis on SFs and TFs, respectively. Figure 6a,b shows the results of SF correlation analysis and TF correlation analysis, respectively. Figure 6a shows that in the 99% confidence interval, nearly 50% of the feature correlations were higher than 0.7, indicating that there was certain multicollinearity between the SFs. Figure 6b shows that within the 99% confidence interval, more than 80% of the feature correlations were above 0.7 or below –0.7, indicating that there was strong multicollinearity between the TFs. Therefore, it was necessary to reduce the dimensionality of the extracted image features.

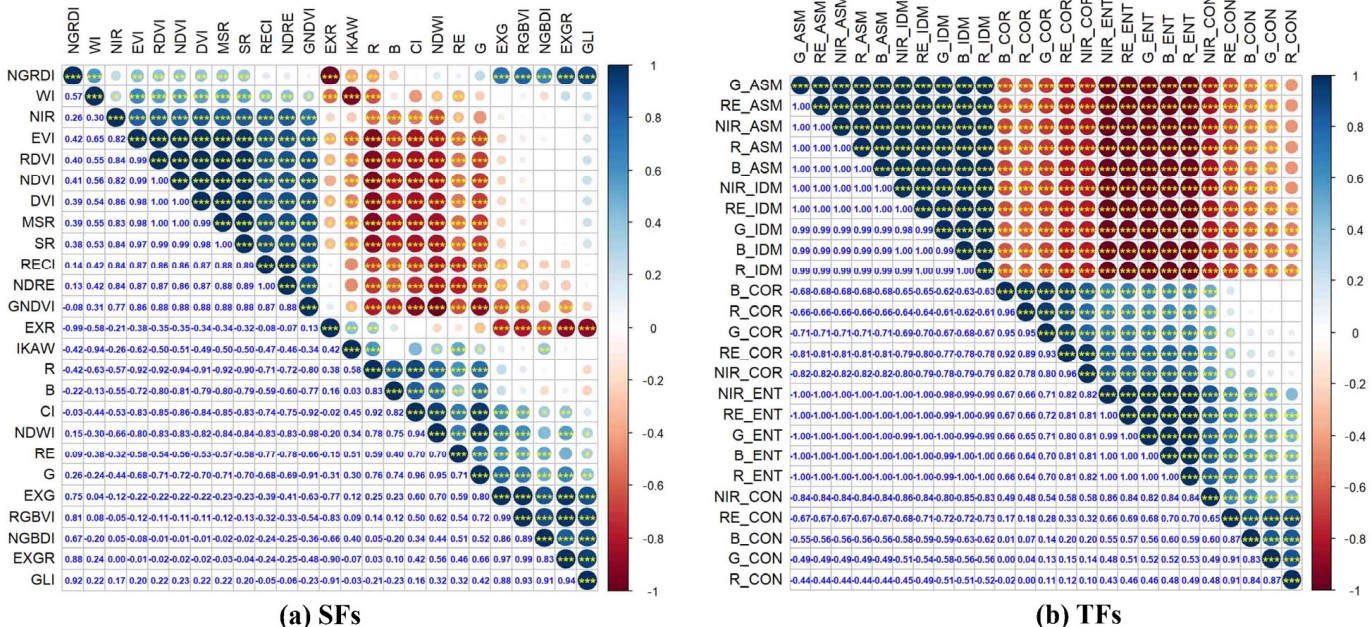

**(a) SFs** **(b) TFs**

**Figure 6.** (**a**) Correlation analysis of SFs; (**b**) correlation analysis of CFs. * stands for 95% confidence level, ** stands for 99% confidence level, and *** stands for 99.9% confidence level.

3.3.2. Selection of the Dimensionality Reduction Algorithm

Figure 7 shows the $R^2$ and *RMSE* values of the RFR_SFs, RFR_TFs, and RFR_CFs models constructed with the low-latitude feature vectors extracted by the three dimensionality reduction algorithms as explanatory variables. Figure 7a,d shows the prediction accuracy of the RFR_SF model. The results showed that LTSA-RFR_SFs had the best fitting effect, the largest $R^2$ of 0.4168, and the smallest *RMSE* of 3.4208%. Figure 7b,e shows the prediction accuracy of the RFR_TF model, in which the $R^2$ of LTSA-RFR_TFs was the largest at 0.4065, and the *RMSE* of LDA-RFR_TFs was the smallest at 3.3646%. However, the $R^2$ of LDA-RFR_TFs was the smallest of the three types of models, and it could be seen that the fitting effect of LTSA-RFR_TFs was the best. Figure 7c,f also shows that the prediction accuracy of the model constructed using CFs through LTSA dimensionality reduction was the highest, the accuracies of LTSA-RFR_SFs and LTSA-RFR_TFs were improved, and the $R^2$ values increased by 15% and 17%, respectively. The *RMSE* values decreased by 5% and 6%, respectively. Figure 7 shows that the model accuracy based on the three dimensionality reduction algorithms was ranked as follows: LTSA-RFR > LDA-RFR > MUV-RFR. Therefore, we used the feature vectors after dimensionality reduction of the LTSA algorithm for model construction.

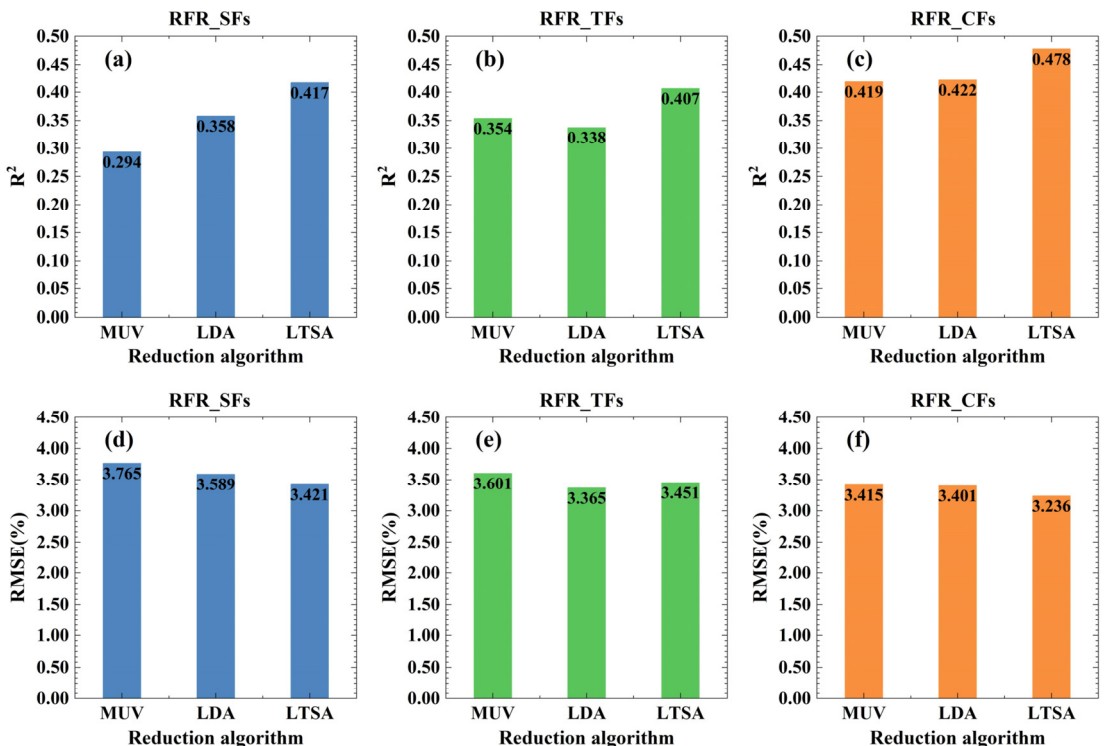

**Figure 7.** RFR-SFs, RFR-TFs, and RFR-CFs model evaluation indicators based on three dimensional reduction algorithms: (**a**,**d**) $R^2$ and RMSE of RFR-SFs model; (**b**,**e**) RFR-TFs $R^2$ and RMSE of the model; (**c**,**f**) $R^2$ and RMSE of the RFR-CFs model.

### 3.3.3. Model Optimization and Verification

Table 5 shows the model evaluation results after applying different algorithms to optimize the numbers of decision trees and node features. The model was evaluated according to the $R^2$, *RMSE*, and *MAPE* values. The goodness of fit of the RFR_SF model was ranked as follows: SSA-RFR_SFs > MBO-RFR_SFs > HHO-RFR_SFs; the goodness of fit of the model constructed by RFR_TFs was ranked as follows: SSA-RFR_TFs > HHO-RFR_TFs > MBO-RFR_TFs; the goodness of fit of the RFR_CF model was consistent with that of the RFR_SF model. The results of the model evaluation showed that the three optimization algorithms could effectively improve the model accuracy, and the optimization effect of SSA was the best. After the optimization of the SSA algorithm, the accuracy of the model constructed based on the three features was significantly improved. Among them, the $R^2$ of RFR_SFs increased by 85%, and the *RMSE* and *MAPE* decreased by 60% and 53%, respectively; the $R^2$ of RFR_TFs increased by 80%, and the *RMSE* and *MAPE* decreased by 50% and 42%, respectively; the $R^2$ of RFR_CFs increased by 73%, and the RMSE and MAPE decreased by 74% and 62%, respectively.

**Table 5.** Comparison of prediction model results for water content of *A. sinensis*.

| Model | $R^2$ | *RMSE*/% | *MAPE*/% | Rank |
|---|---|---|---|---|
| MBO-RFR_SFs | 0.7581 | 2.2030 | 2.5662 | ⑤ |
| HHO-RFR_SFs | 0.7528 | 2.2274 | 2.5388 | ⑥ |
| SSA-RFR_SFs | 0.7731 | 2.1336 | 2.5388 | ④ |
| MBO-RFR_TFs | 0.7265 | 2.3425 | 2.8680 | ⑨ |
| HHO-RFR_TFs | 0.7286 | 2.3338 | 2.8709 | ⑧ |
| SSA-RFR_TFs | 0.7348 | 2.3069 | 2.8412 | ⑦ |
| MBO-RFR_CFs | 0.8204 | 1.8986 | 2.3085 | ② |
| HHO-RFR_CFs | 0.8182 | 1.9101 | 2.3147 | ③ |
| SSA-RFR_CFs | 0.8282 | 1.8566 | 2.2864 | ① |

There were also differences in the model accuracies of RFR_SFs, RFR_CFs, and RFR_SFs. Taking the SSA-RFR with the best goodness of fit as an example, the $R^2$ of SSA-RFR_CFs was 7% and 13% higher than that of SSA-RFR_SFs and SSA-RFR_TFs, respectively, and the *RMSE* was reduced by 15% and 24%, respectively. The $R^2$ of SSA-RFR_SFs was 5% higher than that of SSA-RFR_TFs, and the *RMSE* was reduced by 8%. The comparison of the three types of models showed that the estimation effect of the comprehensive feature was better than that of the single feature, and the accuracy of the model constructed from spectral features was slightly higher than that of the model constructed from texture features, which was consistent with the results in Figure 7c,f.

### 3.3.4. Comparison between SSA-RFR_CFs and CNN_CFs Models

Taking the CFs as an independent variable, a Convolutional Neural Networks (CNN) model for estimating the water content of *A. sinensis* was established and compared with the SSA-RFR_CFs model constructed in this study. Figure 8a,b shows the estimation results of the SSA-RFR_CFs and CNN_CFs models, respectively, where the $R^2$ and *RMSE* of the SSA-RFR_CFs model were 0.8282 and 0.0186, respectively, and the $R^2$ and *RMSE* of the CNN_CFs model were 0.4733 and 0.0325, respectively. The $R^2$ of SSA-RFR_CFS was 75% higher than that of CNN_CFs, while the *RMSE* was 47% lower. At the same time, within the 95% prediction band, the estimated value of SSA-RFR_CFs was more evenly distributed on both sides of the 1:1 line, and the difference between the estimated and measured values was smaller. It could be seen from the above analysis that compared with CNN, the SSA-RFR model proposed in this study had a better fitting effect and higher prediction accuracy.

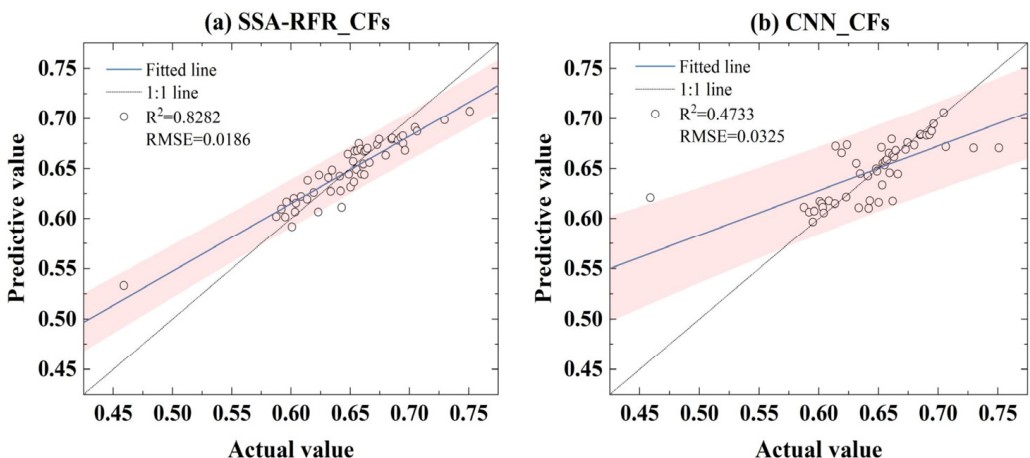

**Figure 8.** SSA-RFR_CFs model (**a**) and CNN_CFs model (**b**) estimation results.

## 4. Discussion

### 4.1. Segmentation of Multispectral Images

Separating the research area from the background is the basis of applying image information to retrieve the plant water content, but some bands of multispectral images are difficult to separate. The multicamera multispectral imager divides the light from the target into several beams and records the information of each band separately. Although the image quality is improved compared with the beam-splitting multispectrometer, it also makes the images of different bands shift, which makes segmentation work difficult [45].

This study provided a way to segment discrete multispectral images. In terms of image registration, in addition to the frequency domain transformation method of FMT, registration methods in the spatial domain, such as image registration methods based on cross-correlation information and feature-based methods, were included [46,47]. In terms of image segmentation, in addition to clustering algorithms such as FLICM, threshold- and vegetation index-based algorithms were also included [48,49]. Therefore, in follow-up re-

search work, image registration and segmentation algorithms can be combined according to specific situations. It should be noted that we used the B-band image as the reference image in this study because of its large foreground–background difference and low noise. However, changing the light source will change the imaging effect of each band. Thus, we must select the reference image according to the quality of the image in practical applications.

### 4.2. Model Construction and Optimization

#### 4.2.1. Extraction of Feature Vectors

In previous studies, some scholars used a single VI to invert the water content of potatoes, and the RVI, which was highly correlated with water content, was considered to be the best predictor [50]. Other scholars used multiple combinations of a single VI to build models separately and chose the best prediction scheme by comparing the performance of different models [13]. The above studies all reduced the dimensionality of explanatory variables by selecting eigenvectors. Although the key factors are retained, this also results in a serious lack of original information. In contrast, this problem does not occur in feature extraction because a kernel function is applied to transform the original information, removing a considerable amount of redundant information on the basis of less information loss [51–53]. Therefore, this study used the LTSA algorithm to extract the eigenvectors of the SFs, TFs, and CFs, using them as explanatory variables to construct an estimation model for the water content of *A. sinensis*. The results showed that the three types of models all showed strong robustness. For example, the MBO-RFR_SFs, HHO-RFR_SFs, and SSA-RFR_SFs models constructed based on the eigenvalues of the SFs in Table 5 all achieved satisfactory prediction results.

#### 4.2.2. Selection of Explanatory Variables

In previous studies, TFs have often been used for classification and biomass prediction [9,54], and there have been relatively few reports on TFs for plant water content prediction. In this study, water conditions had a significant impact on tree height and canopy width, and increases in tree height and canopy width caused the surface texture of *A. sinensis* to constantly change. Therefore, we used TFs as parallel variables of SFs and constructed an *A. sinensis* model based on the TF water content prediction model. The results in Table 5 showed that the prediction model built with TFs as explanatory variables also had strong robustness. However, when we compared the performance of the RFR_TF and RFR_SF models, we found that when the conditions were consistent, the accuracy of the TF-based prediction model was lower than that of the SF-based prediction model. Although this was consistent with the research results of Zhou et al. [18], in our study, the difference was much smaller (5%). This may have been related to the settings of the texture parameters because studies have shown that changing texture parameters can affect the prediction accuracy of texture features [54], and in some studies estimating plant physiological parameters, TFs showed higher prediction ability than SFs [38,55]. The research of Zhang et al. [56] also showed that compared with texture information, spectral information is more susceptible to the influence of instruments and environments. Therefore, TFs, which are complementary to SFs, could be used as another important indicator to predict the water content of *A. sinensis*.

A study on predicting winter rapeseed biomass showed that the fusion of SFs and TFs significantly improved the accuracy of the prediction model [57]. Zhou et al. [18] constructed a wheat stomatal conductance model using comprehensive features and single features, and the results showed that the prediction accuracy of the comprehensive feature model was more than 20% higher than that of the single-feature model. In this study, we constructed a single-feature model and a comprehensive feature model, and the results showed that the predictive ability of the comprehensive feature model was higher than that of the single-feature model, which was consistent with previous research results. Different from previous studies, the accuracy improvement of the comprehensive feature model was not significant compared with the single-feature model in our study. This was because

RFR is an integrated algorithm composed of a large number of sub-models, which has the ability to overcome overfitting. When we used an optimization algorithm to optimize the model parameters, we could maximize the accuracy of the model on the basis of avoiding overfitting of the two types of models.

### 4.2.3. RFR Hyperparameter Optimization

For machine learning models, kernel-based learning methods directly determine the ability of the model to estimate specific variables [58]. When some parameters in the model need to be set manually, it is difficult to ensure the fitting effect of the model. With the development of bionic technology, heuristic algorithms have been applied to the field of model parameter optimization. This is because the heuristic optimization algorithm has strong convergence and generalization capabilities, can take the fitness function as the optimization target, and obtains the optimal solution of the parameters through continuous iteration, thereby improving the fitting effect of the model [59]. However, the update rules of each optimization algorithm are different, and the predicted effects will also be different. For example, in a study on estimating spring corn evapotranspiration, bionic algorithms such as SSA were used to optimize an extreme learning machine (ELM) model, and the results showed that the estimation accuracy of SSA-ELM was significantly higher than that of other models [60]. In this study, MBO, HHO, and SSA were used to optimize the numbers of decision trees and node features of the RFR model, and the results showed that the prediction accuracy of the model was significantly improved by the optimization of the MBO, HHO, and SSA algorithms. Among the three types of algorithms, SSA obtained the best optimization effect, but it was not much different from that of MBO and HHO because the heuristic optimization algorithm is part of a class of probability-based evolutionary algorithms and each algorithm has great similarity in terms of structure and other aspects.

### 4.3. Future Outlook

Water stress is one of the forms of plant adversity and has become the main factor limiting the development of agriculture and forestry. Real-time monitoring of plant water content is of great significance for precision irrigation and is an effective means to address water stress [61]. The image-based estimation method of plant water content has the characteristics of flexibility and strong operability. Extracting the correlation information in an image can indirectly reflect the change in plant water content. In this study, multispectral imaging technology and related machine learning methods were used to construct *A. sinensis* water content prediction models under different water gradients. The $R^2$ of the RFR_CFs model optimized by SSA reached 0.8282, which proved the feasibility of this method in *A. sinensis* water content estimation. Due to the preciousness of *A. sinensis*, we took a small number of samples, but we chose the RFR model as the basic model and trained it using LOOCV, which effectively improved the fitting effect of the model. In future work, we will continue to explore related algorithms in machine learning and deep learning, build a model with higher accuracy and stability, and further improve the accuracy of prediction. In addition to estimating plant water content, multispectral imaging techniques are also being used in other aspects of agroforestry. For example, a rubber blade nitrogen content prediction model was built by obtaining multiple band reflectance data of rubber blades, providing technical support for the rapid detection of rubber blade nitrogen content [62]. There are also studies that have used remote sensing multispectral imaging to classify normal and green bug-infested wheat fields, providing a nondestructive and inexpensive method for pest reconnaissance in the field [63]. In future research, we will continue to explore the application of multispectral imaging technology in nutritional diagnosis and pest control to provide a real-time, nondestructive, and accurate solution for the cultivation and protection of precious tree species.

## 5. Conclusions

In this study, the multi-spectral image of *A. sinensis* was segmented by FFT + FLICM, which provided a new idea for the rapid and accurate segmentation of discrete multi-spectral images. On this basis, we constructed models of *A. sinensis* water content based on image SFs, TFs, and CFs, and analyzed the prediction ability of different image features. In the process of model construction, we used the dimensionality reduction algorithm to extract the feature vectors of the SFs, TFs, and CFs, using them as explanatory variables to effectively avoid overfitting the model. At the same time, we applied the swarm intelligence optimization algorithm to optimize the model parameters and finally determined the best model for predicting the water content of *A. sinensis*. The main conclusions of this study are as follows:

(1) Water treatment has a significant effect on the *A. sinensis* tree height and crown width but little effect on the ground diameter. Compared with waterlogging conditions, drought conditions inhibited the growth of *A. sinensis* more significantly, and a soil water content of 60%–80% was most suitable.

(2) FMT registration can be used to realize the fusion of discrete *A. sinensis* multispectral images, FLICM segmentation effectively suppressed image noise, and the FMT + FLICM scheme realized the fast and accurate segmentation of *A. sinensis* multispectral images.

(3) The effect of TFs on predicting the water content of *A. sinensis* was basically the same as that of SFs, which can be used as another important index to predict the water content of *A. sinensis*. The predictive power of CFs was higher than that of SFs and TFs, but this difference decreased with the optimization of the RFR model.

(4) The model accuracy was greatly improved by optimizing the hyperparameters of the RFR model, and the optimization effect of the SSA algorithm was the best. Compared with the original model, the $R^2$ of SSA-RFR_SFs was improved by 85%, the $R^2$ of SSA-RFR_TFs was improved by 80%, and the $R^2$ of SSA-RFR_CFs was improved by 73%. Among all models, SSA-RFR_CFs had the highest accuracy, and its $R^2$ was 7% and 13% higher than that of SSA-RFR_SFs and SSA-RFR_TFs, respectively.

**Author Contributions:** P.W. performed the experiments, analyzed the data, and wrote the manuscript. X.W. designed the research and conducted the field measurements and collected the samples. Y.W., M.S., X.C. and Y.Y. analyzed the data. All authors have read and agreed to the published version of the manuscript.

**Funding:** This research was funded by the Special Funds for Fundamental Research Business Expenses of the Central Public Welfare Research Institution's "Precise Image Judgment Technology for Health Status of Precious Tree Species", grant number CAFYBB2021ZB002.

**Data Availability Statement:** The data presented in this study are available on request from the corresponding author. The data are not publicly available due to the confidentiality of the project.

**Acknowledgments:** We acknowledge the support from the IFRIT of CAF.

**Conflicts of Interest:** The authors declare no conflict of interest.

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
