# Peer review of "Multispectral Image Determination of Water Content in Aquilaria sinensis Based on Machine Learning"

_forests, doi:10.3390/f14061144_

Round 1

Reviewer 1 Report

This work presents a methodology for the prediction of the water content in Aquilaria sinensis plants, based on the use of multispectral images and random forest regressors.

Both the application domain and the direction of this research are relevant. On the positive side, the problem is well motivated in the introduction section. However, the paper requires a more rigorous scientific structure, and it misses details that need to be incorporated.

Concretely:

-          The use of spectral images and machine learning methods is not new. Moreover, Random Forest is a well-known method in machine learning. Therefore, the actual contribution of the paper in unclear. Please be precise stating the contribution in the introduction section, and highlighting it in both the results and the conclusions.

-          The methodology consists of different steps, whose need is not explained. For instance, why is it needed to perform preprocessing and segmentation instead of computing the features for the whole image? Moreover, why is it needed to manually extract those features instead of learning them through the use of convolutional neural networks, which are the state of the art currently.

-          Including a flow diagram with a paragraph that gives a rough idea of the whole process can help. Then, each subsection can explain each part of the diagram.

-          Line 73 mentions some previous studies. What are those previous studies? Please provide concrete citation and details about those studies.

-          The meaning of the term “absence of intermediate values”, in line 78, is unclear.

-          What does it mean “normal, uniform growth” in line 134?

-          Where do the values of the different parameters listed in section 2.2.1 come from? Why did author use those numbers and not others?

-          Why is one of the groups called control group instead of naming them from 1 to 4?

-          Regarding the camera, as described in line 154, what is the meaning of “same level as the center point”?

-          In relation to the previous observation, are all plants of the same size and width? How do you account for variation with respect to the area of the camera sensor?

-          Explain of acronyms at their first use, i.e., FWHM.

-          Revise the number of images you report. Some times the paper mentions 4, and sometimes 5.

-          Explain the variable a in equation (2). Is it the same as alpha?

-          The column “Formula” of Table 3 must be explained. Also, it is unclear if those formulae are applied pixel wise or over the whole image.

-          Section 2.4 explains only one type of features. What about the other two features? How are they computed?

-          Line 232 says “When the sample size is small, the feature dimension is too high”, but this is not a consequence. It might happen, or not.

-          Section 2.5.1 does not mention how much the dimensionality is reduced. Moreover, why did authors choose those methods? Why not testing the most common methods: PCA and tSNE?

-          Why using only Random Forest? Why not XGBoost or Deep Learning?

-          Line 251 says “as the number of decision trees increases, the RFR model will converge to a lower generalization error”. This is false. In fact, as the number of trees increases, the method might tend to overfit.

-          What does decision tree node stand for in line 256?

-          Why did author incorporated those three optimization methods instead of using the standard random forest methodology? How these methods improve with respect to the standard process?

-          Author mention the use of cross-validation, I can only assume they mean leave-one-out full-cross-validation, as they do not provide more details. Moreover, they do not mention the size of the test set.

-          Also, it is unclear whether the results correspond to the training, validation, or test sets.

-          Section 3.1 must be in the description of the data. Not in the results.

-          The masking step mentioned in line 304 is not explained.

-          Table 4 must be better discussed. It is unclear how these values are further used in the analysis.

-          Why the correlation of section 3.3.1 focusses only on two features, and not on all three of them?

-          The ideas in lines 324 and 325 are inconsistent between them.

-          The description of Figure 5 says that the correlation is between SF and CF, but it is not. It is not a cross-correlation between the two families of features.

-          Results shown in Figure 6 seem to be about the final regression experiment, but then it is unclear how all previous step impact on the final results. What would happen if no pre-processing, segmentation, and dimensionality reduction is performed?

-          What is the meaning of the term “intrinsic dimensions” in line 336?

-           

Other minor observations:

-          Title must be “… water content in Aquilaria sinensis…”

-          First paragraph in page 2, repeats part of its contents.

-          Line 124 mentions that the dry and wet seasons are clear in Figure 1. However, they are not. Figure 1 does not show any type of seasonality.

-          The size of the room, in line 151, must be written more carefully.

-          Be consistent using either the term formula or equation, but not both. Equation is common for scientific papers.

-          All equations must end either with coma or period, depending on whether the next paragraph continues the idea or not.

-          If a paragraph following and equation contains it explanation, then it is not indented. If the equation finished its own idea, and the next paragraph starts a new idea, then the paragraph is indeed indented.

-           

Not many observation, only a few minor improvements that can be made regarging syntax.

Reviewer 2 Report

Dear Sirs,

I appreciated very much the detail you applied regarding methodology of image classification and model construction. Anyway, in my opinion the overall interest of the paper is limited by the fact that models were developed for seedlings in pots, in addition translated in a room for spectral analysis, and implementation of methodology for a desirable application in field plantations is needed. Given the general quality of presentation, I don't undestand why you did not report statistical analysis and/or P values for the effect of soil water content on seedling growth. Given that the subsequent construction is based on this discriminant, it cannot be considered as a minor bias. You can find other minor notes in the attached file.

For this reason, I must ask a major revision.

Best regards

Dear Sirs,

in my opinion English language is of quite good quality. The syntax is clear and the choice of terms is mainly proper. Some minor notes are reported in the attached file.

Round 2

Reviewer 1 Report

Authors have attended some of the comments mande during the first review. But for some reasons some other comments have been neglected.

Moreover, the response letter provides only answers to the observations, but lacks some of the actions taken to improve the document. The document itself has certain changes, but sometimes without the proper accompanying justificartion.

This is not the way to prepare a response letter to a review.

It has been improved. Although some typos still exist.

Reviewer 2 Report

Dears Sirs,

I appreciated your statistical integration of A. sinensis growth differences under various water treatments. Due also to changings requested by the other reviewers, now in my opinion your paper should be accepted for publication, even if its soundness remains limited by the fact that the results must be verified in field conditions.

Best regards

The quality of language in my opinion is quite adequate.
